# PBMNCs Treatment in Critical Limb Ischemia and Candidate Biomarkers of Efficacy

**DOI:** 10.3390/diagnostics12051137

**Published:** 2022-05-04

**Authors:** Matilde Zamboni, Massimo Pedriali, Luca Ferretto, Sabrina Scian, Francesca Ghirardini, Riccardo Bozza, Romeo Martini, Sandro Irsara

**Affiliations:** 1Unit of Vascular and Endovascular Surgery, San Martino Hospital, 32100 Belluno, Italy; wambazamba@icloud.com (M.Z.); francesca.ghirardini@aulss1.veneto.it (F.G.); riccardo.bozza@aulss1.veneto.it (R.B.); romeo.martini@aulss1.veneto.it (R.M.); sandro.irsara@aulss1.veneto.it (S.I.); 2Unit of Surgical Pathology, Azienda Ospedaliera-Universitaria, University of Ferrara, 44121 Ferrara, Italy; mpedriali@gmail.com; 3Unit of Vascular and Endovascular Surgery, Azienda Ospedaliera-Universitaria, University of Ferrara, 44121 Ferrara, Italy; sabrina.scian93@gmail.com

**Keywords:** macrophages, peripheral blood mononuclear cells, critical limb ischemia, biomarkers

## Abstract

When in critical limb ischemia (CLI) the healing process aborts or does not follow an orderly and timely sequence, a chronic vascular wound develops. The latter is major problem today, as their epidemiology is continuously increasing due to the aging population and a growth in the incidence of the underlying diseases. In the US, the mean annualized prevalence of necrotic wounds due to the fact of CLI is 1.33% (95% CI, 1.32–1.34%), and the cost of dressings alone has been estimated at USD 5 billion per year from healthcare budgets. A promising cell treatment in wound healing is the local injection of peripheral blood mononuclear cells (PBMNCs). The treatment is aimed to induce angiogenesis as well to switch inflammatory macrophages, called the M1 phenotype, into anti-inflammatory macrophages, called M2, a phenotype devoted to tissue repair. This mechanism is called polarization and is a critical step for the healing of all human tissues. Regarding the clinical efficacy of PBMNCs, the level of evidence is still low, and a considerable effort is necessary for completing the translational process toward the patient bed site. From this point of view, it is crucial to identify some candidate biomarkers to detect the switching process from M1 to M2 in response to the cell treatment.

## 1. Critical Limb Ischemia: Definition and Scale of the Problem

Peripheral artery disease (PAD) is a chronic and progressive circulation disorder, mainly affecting the lower limbs. In more than 90% of cases, the etiology is atherosclerosis [1], which determines a progressive stenosis of the arteries until their complete occlusion. Vascular trauma and arteritis are less common causes of PAD.

Patients affected by PAD could be completely asymptomatic or present different grades of walking pain, whereas rest pain and ulcerations appear in the advanced stages. Among different classifications based on the ischemic consequences of PAD, the Fontaine [2] and Rutherford [3] scales are the most commonly used in clinical practice and widely cited in the literature. 

The ankle-brachial index (ABI) and the toe-brachial index (TBI) are commonly performed to objectify and stratify patients. The ABI expresses the ratio between systolic arterial pressure at the level of the ankle/brachial artery: a value < 0.9 is diagnostic for PAD, regardless of any symptoms. The ABI is an inexpensive, easy-to-use and widely available method to evaluate PAD, but in cases of extensive calcification of the arteries, it is not accurate and could give false values, especially whether >1. In these conditions, the TBI is more accurate because toe vessels are less susceptible to calcifications. TBI is calculated as the ratio between great toe/brachial pressure, and it is considered normal when >0.6.

Detecting PAD in the early stages is crucial, because it is also a cardiovascular disease marker correlated with survival. PAD affects approximately 202 million people globally. In Europe, it is estimated that 40 million subjects are affected by PAD, a prevalence of approximately 5.3% as referred to by the 750 million Europeans [4]. 

Critical limb ischemia (CLI) is the most advanced stage of PAD, with patients being affected by significant pain, diminished health-related quality of life and, moreover, high mortality and limb loss rates. The epidemiology of CLI is not well known, because only a few population-based studies have been conducted. Nehler et al. reported that the prevalence of CLI in the United States is 1.3%, accounting for 11% of diagnosed PAD cases, among the eligible study population ≥40 years of age [5]. Recently, the Society of Vascular Surgery, in conjunction with the European Society for Vascular Surgery and the World Federation of Vascular Societies, proposed an articulate definition of CLI, expanding it into “chronic limb-threatening ischemia” (CLTI), to stress the frequent epilogue towards limb loss [6]. The authors assert that the diagnosis of CLTI requires objectively documented atherosclerotic PAD associated with ischemic rest pain, ulceration or gangrene. CLTI is clinically characterized by ischemic rest pain typically for more than 2 weeks and/or tissue loss including gangrene and/or non-healing ulceration present for at least 2 weeks. The alterations in hemodynamic parameters include an ABI < 0.4, an absolute ankle pressure < 50 mmHg, an absolute toe pressure (TP) < 30 mmHg, a flat or minimally pulsatile pulse volume recording waveforms and a transcutaneous partial pressure of oxygen (TcPO2) < 30 mmHg [7]. 

The primary treatment of CLI is revascularization, both with surgical and/or endovascular techniques. Recently, Coudene et al. investigated the evolution of limb prognosis in patients with CLI in the ‘Cohorte des patients ARTeriopathes’ (COPART) study [7]. The cohort was divided into two groups according to their inclusion before or after 2011 (total patients of 489 and 450, respectively); the year 2011 was specifically chosen because new recommendations were introduced in terms of secondary prevention and surgical practices. Multivariate analysis showed that the incidence of major amputations (MAs) decreased significantly after 2011 (OR: 1.5 (1.1–2.1)) [8]. This could be explained by the observation that more patients underwent revascularization after 2011, especially with distal angioplasty, which permits to treat patients with many comorbidities, who are often not eligible for surgical bypass. Despite a significant lowering of amputation rates observed in the group after 2011, still approximately 43% of the patients underwent a minor or MA (MA: 17%) [8]. 

Every attempt to restore sufficient direct blood flow to the foot to ensure tissue vitality is conditioned by the status of the distal arteries at the moment of the intervention and by the further progression of the disease. Patency of surgical infragenicular bypasses is directly dependent on the type and quality of the conduit, as the absence of a good great saphenous vein implies poor results. Percutaneous transluminal angioplasty (PTA) is an alternative approach for lower limb revascularization. The mainstay advantages of PTA over a surgical approach are the lower general impact on the patients and the possibility of treating even very distal and small arteries. In addition, PTA often achieves good results in hardly calcified arteries where a surgical anastomosis is not feasible. On the other hand, in complex cases, interventional maneuvers can permanently alter the slender collateral branches of the foot, causing a sudden shift to acute ischemia and a very high probability of MA. Moreover, even if PTA is technically successful, the results in the long term still seem to be inferior compared to a surgical approach, especially if a good vein conduit is available. The major concern about the durability of an endovascular treatment of the lower limb is restenosis, which can occur with and without the implantation of a stent. After a PTA, arteries tend to restenosis which is a phenomenon driven by a lot of mechanisms still under investigations (e.g., inflammation, neointimal hyperplasia and resistance to antimitotic drugs used for coating stents and balloons) [9]. 

Furthermore, the arterial pattern could not be subjected to any procedure as, for example, the so-called “desert foot” [10]. These last conditions are included in the “no-option CLI” patients (NO-CLI), underlining the lack of surgical and/or endovascular revascularization possibilities.

MA is still a treatment widely used in CLI, counting for almost 20% of patients, even if they underwent some kind of revascularization. MA has a dramatic impact on patients, relatives and caregivers’ lives. In addition, MA is very expensive for healthcare systems: different studies in different countries have demonstrated that MA costs more than revascularization with either endovascular or surgical bypass [11,12].

PAD is a global public health issue, and in its final stage (i.e., CLI), it has a dramatic impact on healthcare systems from medical, social and economic points of view, especially when it leads to MA. Recently, a policy statement by the American Heart Association informed about the urgent need to reduce the rate of nontraumatic lower-extremity amputations by 20% by 2030 [13]. Every effort towards this goal must be carried out, and nonconventional treatments play a fundamental role in limiting this increasing burden. Within this setting, over the last decades a growing interest in regenerative medicine can be observed. In particular, some cellular therapies seem to be promising in NO-CLI patients. 

In this review, we analyzed peripheral blood mononuclear cells (PBMNCs), which are demonstrating vascular regeneration capabilities through mechanisms such as angiogenesis, macrophage polarization (from the M1 phenotype to M2) and paracrine stimulation. We summarize the actual evidence of PBMNC application in CLI, suggesting some markers of the efficacy of this therapy, which seems to produce significant clinical improvements.

## 2. Peripheral Blood Mononuclear Cells in CLI

PBMNCs are the mononuclear cells of the blood: lymphocytes (T cells, B cells and NK cells), monocytes and a small fraction of the endothelial progenitor cells (EPCs) [14]. 

Enriched EPCs, usually recognized by CD34+ and CD133+ markers, constitute <0.01% of the PBMNCs and 0.1% of the BMMNCs [14]. 

Lichtenauer et al., in 2011, collected cell culture supernatants obtained by irradiated apoptotic PBMNCs and inoculated them in rat model experimental acute myocardial infarction (AMI) and in a porcine closed-chest reperfused AMI model. The control, magnetic resonance imaging (MRI), showed a reduction in scar tissue, improved cardiac output and a reduction in the extent of infarction [15]. 

This discovery has opened a wide range of research in PBMNCs and regenerative medicine.

The mechanism for how PBMNCs work is still debated. There are three major and known mechanism of action:Angiogenesis;Macrophage polarization from the M1 phenotype to M2;Paracrine stimulation.

Over the last years, several distinct progenitor cell populations have been documented in the PBMNC fraction: hematopoietic stem cells (HSCs), EPCs, mesenchymal stem cells (MSCs), osteoclast and hematopoietic osteoclast precursor cells and a fraction of circulating fibrocytes [16]. PBMNCs are adult stem cells capable of differentiating in vivo and in vitro, according to the characteristics of the site of implantation [14,17].

The inoculation of granulocyte stimulating factor (G-CSF) or granulocyte-macrophage stimulating factor (GM-CSF) is capable, on the one hand, of increasing the production of PBMNCs and CD34+ cells in bone marrow (BM), and on the other, of decreasing the expression of adhesion molecules, permitting an easier migration to peripheral blood [14,18].

G-CSF has been widely used for trials because of the higher number of CD34+ noticed in peripheral blood after stimulation and for the shorter time of apheresis. G-CSF receptors are especially expressed on neutrophils and BM precursor cells [14].

### 2.1. PBMNCs and Angiogenesis

One of the most studied PBMNCs mechanisms of action is angiogenesis. 

The term angiogenesis represents the creation of new blood vessels from pre-existing ones. This phenomenon is distinct from vasculogenesis, which happens during embryogenesis, and is the formation of blood vessels from EPCs and/or angioblasts [19]. 

EPCs are derived from the hemangioblast in BM, which is the precursor of both hematopoietic stem cells and EPCs. The latter could be isolated in BM, in peripheral blood, in adipose tissue and in the umbilical cord, and they constitute 0.001% of the total stem cell population [19]. Their final differentiation is endothelial cells (ECs), contributing to form the inner lining of new blood vessels. Their most common markers are CD34+/VEGFR2+, VEGFR2+/CD 133+ and DiI-Ac-LDL-positive cells [19].

CD34+ is a marker expressed by the HSCs, and flk-1 is a receptor for VEGF expressed by HSCs and EPCs. Both these markers are lost during hematopoietic differentiation and, secondly, at the maturation stage, they are divided into two sub-populations: early and late EPCs. Early EPCs have a more paracrine action, whereas the late EPCs lose CD14+ and are capable of forming colonies of endothelial cells with a high proliferative capacity and, thus, with a high vasculogenic involvement, differentiating into mature ECs and incorporating into new blood vessels [19].

Cells isolated with anti-CD34 or anti-Flk-1 differentiate into ECs in vitro, suggesting that they can contribute, in vivo, to angiogenesis [20].

In diabetes vascular wound, EPCs are dysfunctional because of the hyperglycemia and the higher oxidative stress; in fact, there was found an inverse proportion between HbA1c levels and circulating EPCs [21].

Another proposed mechanism of action of PBMNCs (especially reported in BM mononuclear cells studies) is the stimulation of pericyte differentiation. Pericytes are cells presenting along the capillaries and postcapillary venule walls. Their markers of expression are the platelet growth factor (PDGF) receptor and the proteoglycan NG2 (a coreceptor for PDGF). Usually, they surround arterioles, venules and capillaries in different shapes according to the different kind of smooth muscle fibers. Their principal role is to control blood flow thanks to the intrinsic contractile power of their smooth muscle fibers around capillaries, permitting to adjust the diameter of arterioles and venules [22]. This capacity, whether improved, could potentiate the tissue vascularization.

Moryia et al. retrospectively studied patients with CLI treated with PBMNCs, noticing that, clinically, ischemic rest pain ameliorates after this kind of treatment, but they also noticed a wide range of clinical responses. They investigated the peak plasma level of VEGF after PBMNC injection, discovering a significantly higher expression in responder patients than in non-responders. In particular, the majority of non-responders were patients with multiple comorbidities, especially chronic renal insufficiency in hemodialysis treatment [23]. 

It has been demonstrated that monocytes and macrophages maintain their angiogenic potency in diabetic patients, while HSCs showed a reduced angiogenic power [24]. In addition, Spaltro et al. tested PBMNC injection into a mouse model of hind limb ischemia, finding an induced tissue neo-vascularization by an increasing number of capillaries, arterioles and regenerative fibers [25]. These data were confirmed by De Angelis et al. after PBMNC implantation in NO-CLI patients including a subset of diabetic patients [25]. The histological data confirmed the formation of dermal granulation tissue, an increased monocytes tissue concentration and newly formed microvessels, whereas dermal inflammation and monocyte infiltration were reduced [26].

Moreover, PBMNC implantation promotes other significant changes in the diabetic foot tissues such as the inhibition of HIF-1, NF-KB and TNF-alpha; an increase of VEGF; the appearance of newly formed capillaries [27].

### 2.2. Macrophages Polarization from the M1 Phenotype to M2

#### 2.2.1. Wound Healing and the Paradigm of M1–M2 Macrophages

Physiologically, adult wound healing is a process characterized by three independent overlapping phases: inflammation, granulation and remodeling.

During inflammation, polymorphonuclear neutrophils are the first cells infiltrating the damaged tissue, followed by circulating monocytes that, once achieved at the wound site, differentiate into macrophages [28].

Macrophages represent the most important protagonist during wound repair. They are extremely plastic and dynamic, and they are capable of changing their phenotype according to different external stimuli [29,30]. In particular, the microbial components lipopolysaccharide (LPS), Toll-like receptor (TLR) ligands and interferon-gamma (IFN-γ) usually activate M1 macrophages, whereas IL4/IL-13, immune complexes and TLR, IL-1 receptor ligands and IL-10 stimulate the alternative M2 activation. This phenomenon is the so-called macrophage polarization [31,32].

During inflammation, the M1 macrophages support microbicidal and cytotoxic host defense functions by releasing high concentrations of pro-inflammatory cytokines such as tumor necrosis factor α (TNFα), interleukin (IL)-1β, IL-6 and IL-12 and also reactive oxygen species (ROS) [33,34]. The accumulation of apoptotic cells stimulates the switch from the M1 to the M2 phenotype. From this moment on, the anti-inflammatory or granulation one phases begin, characterized by anti-inflammatory cytokines (IL-10); multiple growth factors such as transforming growth factor (TGF)-β, vascular endothelial growth factor (VEGF), platelet-derived growth factor (PDGF); the deposition of extracellular matrix (ECM) directed by fibroblasts and myofibroblasts. In parallel, M2 macrophages and activated fibroblasts also release proangiogenic factors, recruiting EPCs and improving new vessels formation [35,36].

The origin of macrophages has long been debated over the last years [37]. Nowadays, it is well known that the majority of tissue macrophages are almost already present in the target tissues, even before the definitive hematopoiesis is performed. Monocytes derived from a common progenitor, called a macrophage dendritic cell precursor (MDP), are able to differentiate both inflammatory macrophages and dendritic cells [38]. MDPs can also differentiate into other hematopoietic lineages.

Pathology is frequently associated with dynamic changes in macrophages activation. Classically, activation of M1 or M1-like cells is implicated in initiating and sustaining inflammation, and the activation of M2 or M2-like cells is associated with resolution or smoldering of chronic inflammation [39].

Plasticity is one of the major characteristics of monocytes and macrophages and the M1–M2 polarization is not yet totally defined. It has also been hypothesized the existence of a third macrophages phenotype: M3 or switching phenotype which could be capable to direct the M1/M2 polarization by inducing the secretion of M1/M2 activators [40].

An M1/M2 discriminating system is critically needed to improve the knowledge about the macrophages phenotype and their potential in diagnostic and therapeutic use [41,42]. Unfortunately, it is quite difficult detect specific M1/M2 markers, both in vivo and in vitro, and no pure M1–M2 macrophage marker has yet been found [40]. It seems that M1 macrophages express M2 markers and vice versa. Moreover, in some inflammatory cases, M1-like or M2-like phenotypes could have specific markers expressed by both phenotypes.

Historically, the first studies on M1/M2 markers were practiced in mice with the help of complementary DNA (cDNA) subtraction [39,41] followed by human macrophage transcriptome profiling [43,44,45,46].

Jablonski et al. studied M1/M2 gene expression during their different activations using murine in vitro samples. Specifically, they performed different transcriptional messenger RNA (mRNA) and applied them to undifferentiated (M0), M1 and M2 murine macrophages, showing that M1 and M2 macrophages co-express many genes such as the transcription factors (TFs): Kruppel-like factor (Klf) 4 and activating transcription factor (Atf) 4. However, they also identified M1 and M2 specific genes: CD38, G protein-coupled receptor 18 (Gpr18) and formyl peptide receptor 2 (Fpr2) expressed by the M1 population and early-growth response gene 2 (Egr2) and c-myelocytomatosis oncogene product (c-Myc) expressed by the M2 macrophages [47].

Figure 1 shows macrophage gene expression comparing M1 vs. M0 (Figure 1a) and M2 vs. M0 (Figure 1b). The M0 macrophage population are quiescent nonactivated cells that, according to the environmental stimuli, can change their phenotype into M1 or M2 macrophages.

Jablonski et al. noticed during M1 activation an increased expression of 629 genes and a decreased expression of 732 genes, whereas M2 activation was characterized by 388 upregulation genes. Furthermore, the study group compared their results with the already known mouse and human M1 markers finding 21 M1 common markers such as nitric oxide synthase 2 (Nos2), IL-1β, IL-6, IL-12β, CC chemokine Receptor 7 (CCR7), inhibin beta A (Inhba) and tumor necrosis factor α (TNF-α). The same study was made on M2 markers finding the same markers in Arginase 1 (Arg1), Chitinase 3-like-3 (Chi3l3/Ym1), Resistin-like molecule alpha (Retnla/Fizz1), Egr2, fibronectin 1 (Fn1) and mannose receptor C-type 1 (Mrc1/CD206) [47].

Figure 2 shows genes up- and/or downregulated by both M1 and M2 macrophages (upper-right and bottom-left quadrants). Distinct activated M1 or M2 genes are reported in the bottom-right and upper-left quadrants. These two sets of genes provide a promising group of M1 and M2 macrophage specific markers that may be used to distinguish these two populations during clinical practice [47,48,49,50].

#### 2.2.2. PBMNCs and Macrophages Polarization

In PAD and, especially, in diabetic PAD patients, macrophages are mostly stopped into the M1 phenotype. It seems that in diabetic chronic wounds, the persistent activation of M1 macrophages induces a higher expression of pro-inflammatory cytokines, inducible NO synthase and metalloproteinase 9, perpetuating an inflammatory state and impairing new granulation tissue formation. The macrophages’ deficiency to switch from the M1 to M2 phenotype could be attributed to their impaired phagocytosis of apoptotic cells in the diabetic microenvironment [51].

Furthermore, macrophages found in chronic wounds showed a reduce ability to eliminate dead neutrophils. This process could cause the creation of a highly inflammatory state with an excess of inflammatory molecules such as TNF-α and IL-1β [52,53].

The consequent cellular and biochemical adaptations after PBMNC implantation favor the establishment of conditions similar to the physiological ones and progressively support the regeneration of damaged tissues and wound healing. This phenomenon has been measured biochemically as the inhibition of HIF, NF-KB and TNF-alpha; in the progressive polarization of M1 into M2 macrophages; the increase in VEGF and newly formed capillaries [27].

Moreover, by means of the histological examination of incisional biopsies after PBMNC treatment, it has been noticed that the perilesional area of diabetic nonhealing wounds had a more powerful polarization of M1 macrophages into the M2 phenotype [51].

Masuda et al. studied a method to obtain a quality- and quantity-control (QQ) culture of EPCs, assuming the paucity of stem cells in PBMNCs. The polarization of monocytes/macrophages from the M1 phenotype to M2 is usually marked by an increase in CD 206+ cells and a decrease in CCR2+ cells. In particular, they noticed that monocyte/macrophages in the QQ cultures have a higher tendency to the angiogenic and anti-inflammatory phenotype, improving the regenerative process during ischemia [52].

They also focalized their studies on the lymphocyte lineage cells during regeneration, noticing that B lymphocytes, NK cells and cytotoxic T cells significantly decrease or disappear during the regenerative phase, whereas helper T cells are the last disappearing lymphocyte population [54].

Over the last years, the interaction between monocytes/macrophages and T lymphocytes has been studied, discovering that IFN-c, produced by Th1 lymphocytes, induces the M1 phenotype, whereas IL-4, IL-13 and IL-10, produced by Th2 and regulatory T lymphocytes, induce the alternative activation of M2 macrophages. Moreover, M1 macrophages activate Th1 lymphocytes, secreting IL-12 and IL-6, whereas M2 macrophages improve Th2 and regulatory T-lymphocyte functions, producing IL-10 and TGF-b [54]. Finally, the interactions between M2 macrophages, Th2 lymphocytes and regulatory T cells improve and accelerate the angiogenic and anti-inflammatory phases in the QQ cultures [54].

### 2.3. PBMNCs and Paracrine Stimulation

PBMNCs release pleiotropic paracrine factors stimulating tissue regeneration, and maybe this is one of their strongest ways of contributing to angiogenesis [55].

Thum et al. proposed the idea that the anti-inflammatory response to stem cells depends on the fact that the 5–25% of the injected stem cells are apoptotic and, physiologically, apoptotic neutrophils switch the inflammatory response to the anti-inflammatory one, secreting TGF-β, PGE2 and inhibiting inflammatory mediators. It was demonstrated that the injection of paracrine factors released by apoptotic PBMNCs eases the myocardial damage and contributes the preservation of the myocardial function and microvascular perfusion [15,56].

A few studies have also demonstrated that the injection of PBMNCs is strongly associated with the secretion of paracrine and pro-angiogenic factors such as b-FGF, VEGF, HGF and angiopoietin-1 [37].

Rehman et al. analyzed EPC proliferation and surface marker expression by means of flow cytometry, finding that cultured EPCs upregulated monocyte activation and macrophage differentiation markers and secreted numerous growth factors such as VEGF, HGF, GCSF and GMCSF [57].

These results express the powerful way of PMNBCs and, especially, of monocytes to contribute to angiogenesis and wound regeneration thanks to their paracrine stimulation. Specifically, it was noticed that, especially in diabetic CLI patients, the CD14+ monocytes seemed to react much more to the hypoxic stimuli and had better paracrine action to stimulate angiogenesis in respect to CD34+ cells [58].

Recently, a study on the secretome of stressed PBMNCs was published. It was found that they improve short- and long-term cardiac performance in a porcine infarction model. The transcriptional analyses of the nonperfused and perfused heart 24 h after myocardial infarction showed a highly tissue-specific effect of the secretome and, except for the transition zone, a uniform downregulation of pro-inflammatory factors and pathways [57]. Simultaneously, the secretome strongly promoted the expression of genes that are essential for heart functioning in the nonperfused area [59].

Many studies on the secretome of γ-irradiated PBMCs have found that they are able to attenuate the hypoxia-induced cell damage in AMI and CLI, accelerating wound healing in a diabetic mouse model [60].

Moreover, it was discovered that the use of particular filtration systems allows PBMNCs to migrate in response to a certain gradient of VEGF and stromal-derived factor 1 “SDF-1”. Interestingly, filtration preserves and optimizes the release of paracrine factors, whereas it is significantly reduced when the cell concentrate is produced by centrifugation [25].

## 3. Clinical Experience of PBMNCs in CLI

Over the last years, a few early phases of randomized clinical trials (RCTs) on adults have demonstrated that cell-based therapy for vascular regenerations in patients with CLI or NO-CLI is a safe, feasible and effective procedure [61,62,63,64,65,66,67,68].

### 3.1. PBMNCs in NO-CLI Patients

In 2002, Tateishi-Yuyama et al. published a pilot study RCT that compared the use of PBMNCs vs. BMMNCs in the same patients. In particular, they observed the different implantations inoculated in both legs in patients with bilateral CLI. The comparison demonstrated similar results in legs inoculated with PBMNCs and in those inoculated with BMMNCs, especially in the improvement of collateral vessels. Clinically, patients had an improvement in the ABI, a rest pain reduction/resolving and an amelioration in pain-free walking time [69].

Obviously, the availability and the extraction of peripheral blood is easier, safer and painless in respect to the BM. In 2005, Kawamura et al. published a study in which they treated 87 CLI patients with PBMNCs extracted by means of apheresis after G-CSF subcutaneous inoculation, incubated and implanted intramuscularly. The majority of patients had diabetes and chronic renal insufficiency, but the preliminary studies showed an improvement in VEGF, CD34+ and Factor VIII as well as a clinical amelioration of the patients [70].

The next year, Tateno et al., after several studies on mice, treated 29 patients with NO-CLI, the majority of whom had the indication to MA. PBMNCs were inoculated into the ischemic muscles two times within a month. After one year, the majority showed an improvement in ulcer healing and rest pain and a decrease in MA. After treatment, they noticed that the peak levels of plasma IL-1β, IL-6 and VEGF were markedly higher in responders than in non-responders [71].

In 2007, Hoshino et al. treated seven diabetic patients in hemodialysis (HD) with PBMNCs, and they noticed an improvement in quality of life at 24 weeks, without major events. They also considered the use of peripheral blood in these patients, because people in HD mostly had important comorbidities and, thus, general anesthesia could be too invasive and often the BM could be hypocellular [72].

In 2009, Moriya et al. worked retrospectively on 42 patients with CLI, treated from 2002 to 2005 with PBMNCs concentrated after apheresis and inoculated intramuscularly, as shown in the previous studies. They observed a significant decrease in rest pain of >70%, an improvement in walking distance and ulcer healing and the MA rate was <10% [23].

Looking at clinical studies on PBMNCs from 2005 until today, seven RCTs were conducted comparing the efficacy of PBMNCs to placebo or non-placebo controlled. The main characteristics of the studies are shown in Table 1. All the PBMNCs were autologous and administrated intramuscularly. The extraction was made, in the majority, as mentioned before, by means of apheresis after pretreatment with subcutaneous G-CSF. The preparation was then incubated, centrifugated and concentrated [61].

The primary outcomes, in all of the RCTs, were the reduction in MA, the increase in complete ulcer healing, the improvement in ABI and the reduction in all-cause mortality. They all showed promising results, but the paucity of patients, the short FUs and the cessation at early phases must be improved with a multicentric phase 3 RCTs.

Pan et al., in a meta-analysis on CD 34+, demonstrated that these cells are associated to a significant and higher rate of complete ulcer healing and a lower amputation rate in respect to controls and, especially, the high dose CD34+ in respect to the lower ones [73].

The limitations of these studies are, firstly, the sample size, which are usually very low, and, secondly, the paucity of follow up.

Another limitation is the heterogeneity of the samples, in both cases and controls, which does not allow for comparison of the RCTs and to obtain more significative results [74].

### 3.2. PBMNCs in Revascularized but Nonhealing Patients

Frequently, revascularized patients with both below-the-knee (btk) surgical by-pass and PTA do not heal. Okazaki et al. observed that the percentage of people healed after treatment was 56.3%, 63.4% and 64% at 1, 2 and 3 years [75].

Treatment with PBMNCs could be also a good choice as an adjuvant therapy, especially in diabetic patient with microangiopathy or patients with just one patent btk.

## 4. Candidate Biomarkers and Future Perspective

Such innovative cellular therapies require research that can allow for the effective induction of a cell regeneration process to be monitored, hand in hand with the extinction of the chronicity of the inflammatory process. These are still extremely innovative and non-widespread therapies, but they are truly intriguing and promising. At least, the identification of biomarkers becomes essential to be able to easily monitor the macrophage phenotype switch and to improve the timing of subsequent cell graft treatments. In fact, in clinical practice, at least 2–3 consecutive treatments, on average, are required to reach the results achieved in the above cited clinical trials. For this reason, it would be very important to be able to identify the timing of the treatments that must overlap the first induction of the cellular signaling of tissue repair. In this phase of the research in the field, it would be absolutely premature to think about using circulating and easily doseable markers in the serum. However, taking into consideration the simplicity of a vascular wound biopsy could be proposed to study biomarkers that are part of the pathways activated by the switching process of macrophages from M1 to M2. From this point of view, the use of sample biopsies easily taken from the ulcerated area appears extremely promising. Subsequently, immunohistochemistry could allow for the detection of exclusive biomarkers.

There should be the creation of a research lab platform to study the biological processes happening after PBMNC injection, which kind of biomarkers are expressed in blood and tissues via macrophage polarization from the M1 to M2 phenotype and which genes are involved during the tissue repair. Particularly, an investigation would be focused on the so-called “vitagenes”, a group of genes strictly involved in the preservation of cellular homeostasis during stressful conditions. The latter include, heat shock proteins (Hsp32 and Hsp70), heme oxygenase-1, the thioredoxin system and the sirtuin system [76,77].

Another promising biomarker that needs to be investigated is c-Myc.

The c-Myc transcription factor (c-MYC in humans) is a proto-oncogene involved, directly or by regulating different proteins expression, in various cellular processes. It controls many cellular functions and metabolism such as proliferation, tissue remodeling, angiogenesis, apoptosis, cell survival and the production of inflammatory or anti-inflammatory cytokines [78,79]. C-MYC is expressed in vitro by human macrophages M2 and in vivo in certain types of human tumor-associated macrophages characterized by a M2-like hyperactivation status [58,80].

It is extremely important, noting that human studies in vitro on macrophages showed that c-MYC expression is restricted to the M2 phenotype and is almost undetectable in M0 and M1 macrophages, whereas murine M0 macrophages express small but detectable levels of c-Myc [47,80].

Figure 3 represents three different histologic preparations obtained by samples of a nonhealing chronic vascular wound at one week after PBMNC treatment. CD68 is a pan-macrophages marker expressed in every one of the three different phenotypes (Figure 3b). Differently, c-Myc seems to be expressed, in humans, only by M2 macrophages. The PBMNC treatment evidently induced the macrophages polarization. C-myc is also abundantly and widely found during optical microscopy examinations. Therefore, it could be a strong biomarker and a potential candidate to follow up the pathophysiological changes induced by the cellular treatment. Further studies which correlate the reparative process with the expression of c-Myc focusing also on its function on macrophages are, of course, warranted.

## 5. Conclusions

PBMNC treatment is a promising treatment in patients with CLI, both revascularized and NO-CLI. However, there is the need to increase the level of evidence, elucidate the mechanism of their action in hypo-oxygenated tissues and determine prognostic biomarkers, firstly, expressed by peripheral tissues and, secondly, by blood. Biomarkers of efficacy could improve our knowledge on PBMNCs and permit to distinguish responder and non-responder patients and ameliorate PBMNC clinical application.

## Figures and Tables

**Figure 1 diagnostics-12-01137-f001:**
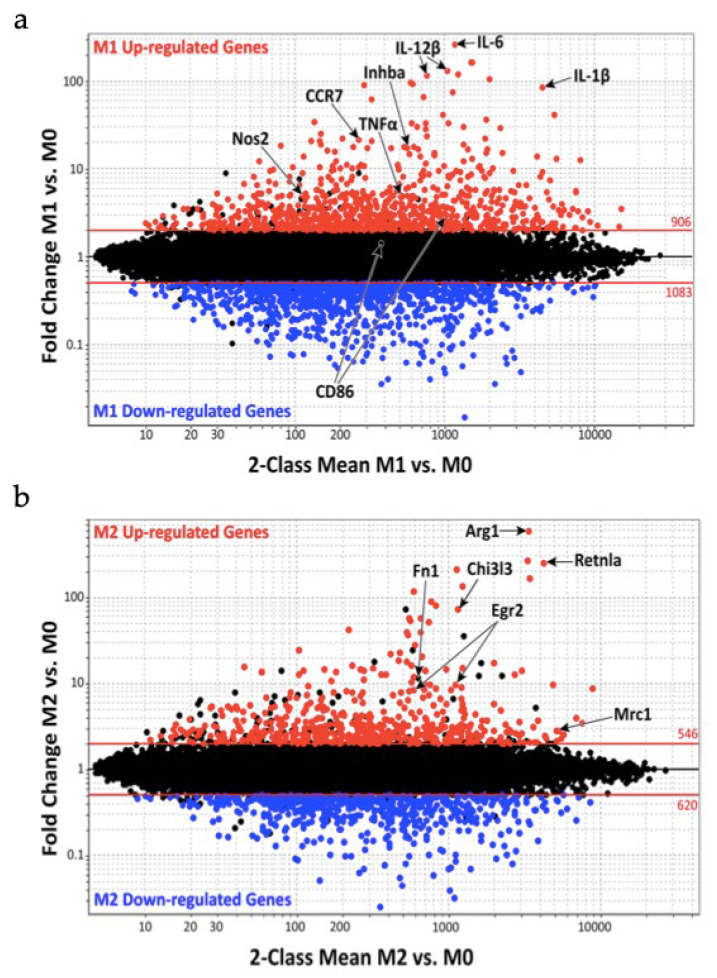
Macrophage gene expression during the classically activated M1 and alternatively activated M2 phenotypes: (**a**) genes up and down regulation during M1 activation; (**b**) genes up and down regulation during M2 activation.

**Figure 2 diagnostics-12-01137-f002:**
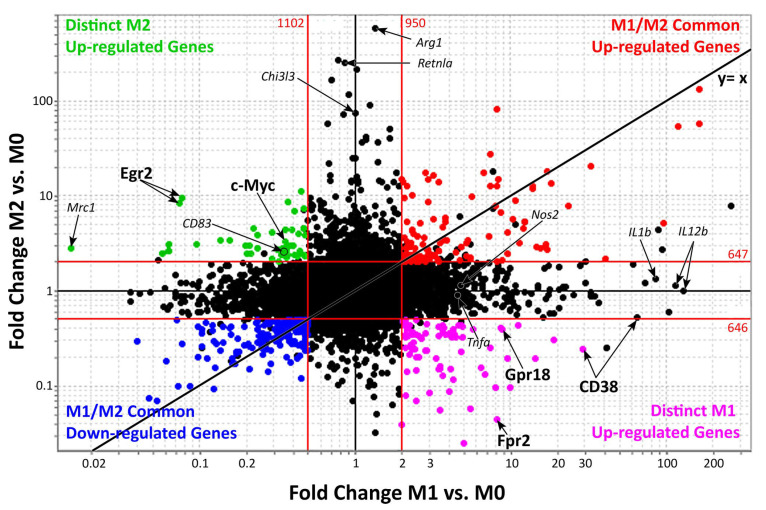
Genes expressed during M1 and M2 macrophages activation.

**Figure 3 diagnostics-12-01137-f003:**
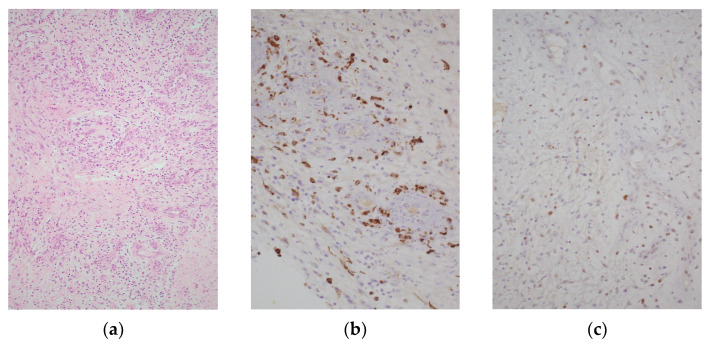
Bioptic human tissue after treatment with PBMNCs: (**a**) hematoxylin and eosin—granulation tissue permeated by chronic inflammation with a rich granulocyte and macrophage population; (**b**) immunohistochemistry (IHC)—CD68 is a pan macrophage marker (M0 + M1 + M2); (**c**) IHC—C-Myc marks the M2 subpopulation in vivo.

**Table 1 diagnostics-12-01137-t001:** Principal characteristics of RCT studies on PBMNCs.

Author	Year	RCT Design	PBMNC Culture	Cases (*n*)	Controls (*n*)	Follow Up (Months)
Huang	2005	Prospective	G-CSF	PBMNC (14)	Standard care + prostaglandin E1 (14)	3
Losordo	2012	Prospective Double Blinded	G-CSF	PBMNC (16)	Standard care + saline + blood (12)	12
Ozturk	2012	Prospective Open Label	G-CSF	PBMNC (20)	Standard care (20)	4
Mohammadzadeh	2013	Prospective	G-CSF	PBMNC (7)	Standard care + saline (14)	3
Szabo	2013	Prospective Open Label		PBMNC (10)	Standard care (10)	3
Raval	2014	Prospective Double Blinded	G-CSF	PBMNC (3)	Standard care + saline (3)	12
Dong	2018	Prospective Single Blinded Non-Inferiority	G-CSF	CD34+ (25)	PBMNC (25)	24

## Data Availability

The data presented in this study are available in the article.

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
