# Peer review of "PBMNCs Treatment in Critical Limb Ischemia and Candidate Biomarkers of Efficacy"

_diagnostics, 2022, doi:10.3390/diagnostics12051137_

Round 1

Reviewer 1 Report

The authors discussed the role of PBMNCs Treatment in Critical Limb Ischemia. The topic is interesting however, there is no clear link between CLI and different parameters discussed. The manuscript should be rewritten to be easier to follow by readers.  

  1. Critical Limb Ischemia: definition and scale of the problem: Add part on relation between CLI and PBMNCs
  2. The paradigm of M1-M2 macrophages: No discussion about M1-M2 macrophages and CLI. Part should be under section 3 (Peripheral Blood Mononuclear Cells in CLI)
  3. Peripheral Blood Mononuclear Cells in CLI: no discussion about PBMNCs and CLI
    • Angiogenesis: no discussion about PBMNCs and CLI
    • Macrophages Polarization from M1 phenotype to M2: no discussion about studies on macrophages and CLI
    • Paracrine stimulation: no discussion about studies Paracrine stimulation and CLI

Reviewer 2 Report

Zamboni et al. prepared a review paper related to an important subject such as critical limb ischemia (CLI), one of the manifestations of cardiovascular diseases (CVDs). CVD is the most important cause of morbidity and mortality worldwide. Moreover, CLI is a serious, life-threatening clinical condition that can lead to sepsis and the necessity of limb amputation, which is associated with worsening quality of life. So, research on new possibilities in CLI treatment is fundamental. The use of peripheral blood mononuclear cells seems to be a promising direction.

Generally, the paper is quite interesting and may be considered for publication in “Diagnostics” in the future, but some significant revisions are required.

I’m afraid I have to disagree with the definition of CLI formulated by the Authors in the first sentence of Chapter 1. CLI is not a complication but the most severe form of peripheral arterial disease. I’m not sure whether it can be said that CLI is frequently developed in patients with PAD. The definition of PAD is the value of the ankle-brachial index below 0.9. In the population with PAD, some cases are without clinical symptoms. Only some people with PAD develop CLI. On the other hand, in the other part of Chapter 1, the Authors define CLI one more time, writing that CLI is a complication of PAD. So, Chapter 1 must be changed and written in a more structured form. Chapter 1 is too long and too detailed to describe different concepts. It should contain basic information on PAD, its definition, and epidemiology. The scales of Fontaine and Rutherford should be mentioned, ABI and TBI should be defined and shortly described, and the definition and epidemiology of CLI should be shortly described. Moreover, it should be said that revascularization, surgical and percutaneous, plays a vital role in the treatment of CLI, although restenosis is a serious problem that diminishes its effectiveness, as well as the most important mechanism taking part in the process of restenosis should be mentioned (additional references can be used, see for example, here DOI: 10.3390/ijerph17249339; DOI: 10.3390/ijerph182211970; DOI: 10.23736/S0392-9590.18.03996-2; DOI: 10.1016/j.ccl.2021.06.006; DOI: 10.1007/s11886-022-01643-2; DOI: 10.23736/S0021-9509.21.11725-2).

In my opinion, chapter 2 on macrophage subpopulations is very long and should be limited only to such information that, in the Authors' opinion, is necessary and important in the context of the leading topic of the work.

In my opinion, the title of the work should be rethought by the Authors. The work relates to the basic knowledge of CLI (Chapter 1), general information about macrophages and their subpopulations (Chapter 2), the therapeutic mechanisms of cellular therapies in CLI (Chapter 3), and the clinical application of cell therapies in CLI (Chapter 4), and a short chapter showing prospects and suggesting the use of specific biomarkers (Chapter 5). Thus, the issue of biomarkers is only a tiny part of the work. Moreover, a clear summary of what biomarkers can be used in clinical practice is needed.

It should be, for example, “Masuda et al.” (lack of the point) (line 332 and others)

A reference number should be given before a point, for example, “[7].”, not “. [7]”.

The English language is at a good level, although it should still be checked and corrected by a linguist.

Round 2

Reviewer 1 Report

Matilde et al. reported a review study on PBMNCs role in critical limb ischemia. They focused on the role of macrophages polarization and CLI potential treatment. The topic is of interest, as it discuss the implication of immune system and more specifically the innate immune part as a key inflammatory moderator of inflammation and its interaction with CLI.

Plasticity is a major characteristic of Monocyte/Macrophages. Authors should mention clearly that polarization of macrophage is still not yet totally defined as M1, M2, M3 …   however we have Like-M1 or Like-M2 phenotype depend on expression of specific markers that may be expressed in both phenotype for some inflammatory cases.

  • Typo error should be revised
  • Add reference Line 38 (after “atherosclerosis’)
  • Line 158 add Ref

Reviewer 2 Report

The paper has been significantly improved and currently it can be considered for publication.

Author Response

We especially thank the Reviewer for the great improvements suggested.